

**WRF-ELM v1.0: a Regional Climate Model to Study Atmosphere-Land Interactions Over**
**Heterogeneous Land Use Regions**
Huilin Huang[1*], Yun Qian[1*], Gautam Bisht[1], Jiali Wang[2], Tirthankar Chakraborty[1], Dalei Hao[1], Jianfeng Li[1],
Travis Thurber[1], Balwainder Singh[1], Zhao Yang[1], Ye Liu[1], Pengfei Xue[2,3], William J. Sacks[4], Ethan Coon[5],
and Robert Hetland[1]
1. Atmospheric, Climate, and Earth Sciences Division, Pacific Northwest National Laboratory, Richland,
WA, USA
2. Environmental Science Division, Argonne National Laboratory, Lemont, IL, USA.
3. Great Lakes Research Center, Michigan Technology University, Houghton, MI, USA.
4. Climate & Global Dynamics Lab, NSF National Center for Atmospheric Research, CO, USA
5. Climate Change Science Institute, Oak Ridge National Laboratory, TN, USA
Corresponding to: Huilin Huang (huilin.huang@pnnl.gov) and Yun Qian (yun.qian@pnnl.gov)





**Abstract**

23        The Energy Exascale Earth System Model (E3SM) Land Model (ELM) is a state-of-the-art land

surface model that simulates the intricate interactions between the terrestrial land surface and other
components of the Earth system. Originating from the Community Land Model (CLM) version 4.5, ELM
has been under active development, with added new features and functionality, including plant hydraulics,
radiation-topography interaction, subsurface multiphase flow, and more explicit land use and management
practices. This study integrates ELM v2.1 with the Weather Research and Forecasting (WRF) Model
through a modified Lightweight Infrastructure for Land Atmosphere Coupling (LILAC) framework,
enabling affordable high-resolution regional modeling by leveraging ELM's innovative features alongside
WRF's diverse atmospheric parameterization options. This framework includes a top-level driver for
variable communication between WRF and ELM and Earth System Modeling Framework (ESMF) caps for
WRF atmospheric component and ELM workflow control, encompassing initialization, execution, and
finalization. Importantly, this LILAC-ESMF framework demonstrates a more modular approach compared
to previous coupling efforts between WRF and land surface models. It maintains the integrity of the ELM's
source code structure and facilitates the transfer of future developments in ELM to WRF-ELM.

To test the ability of the coupled model in capturing land-atmosphere interactions over regions with

a variety of land uses and land covers, we conducted high-resolution (4 km) WRF-ELM ensemble
simulations over the Great Lakes Region (GLR) in the summer of 2018 and systematically compared the
results against observations, reanalysis data, and WRF-CTSM (WRF-coupled with the Community
Terrestrial Systems Model). In general, the coupled WRF-ELM model has reasonably captured the spatial
distribution of surface state variables and fluxes across the GLR, particularly over the natural vegetation
areas. The evaluation results provide a baseline reference for further improvements of ELM in the regional
application of high-resolution weather and climate predictions. Our work serves as an example to the model
development community for expanding an advanced land surface model's capability to represent fully-
coupled land-atmosphere interactions at fine spatial scales. The development and release of WRF-ELM
marks a significant advancement for the ELM user community, providing opportunities for fine-scale



regional representation, parameter calibration in coupled mode, and examination of new schemes with
atmospheric feedback.



## 1. Introduction

Land surface models (LSMs) solve the exchange of water, energy, and carbon fluxes between the land surface and atmosphere (Fisher and Koven, 2020), and are frequently used to simulate response of the Earth's surface to both anthropogenic and natural forcings (Best et al., 2015). These models describe biogeophysical properties like surface roughness, albedo, and evapotranspiration efficiency, characteristics crucial for modeling the land's influence on meteorological processes (Xue et al., 1991; Dai et al., 2003; Dickinson, 1984; Sellers et al., 1986). Originally developed to support weather and climate modeling, LSMs were designed to provide essential lower boundary conditions such as radiation, energy, and water fluxes to the atmosphere.

Over time, LSMs have evolved significantly, with representations of increasingly complex processes that impact land surface dynamics and belowground processes, with their feedback to the atmosphere being incrementally added in newer-generation LSMs. As a consequence of all these advancements, the applicability and scope of LSMs has broadened substantially from their initial versions, introducing sophisticated representations of plant hydraulics (Fang et al., 2022; Xu et al., 2023), wildfire (Thonicke et al., 2010; Li et al., 2012; Huang et al., 2020a; Huang et al., 2021), soil biogeochemistry and nutrient cycling (Li et al., 1992; Parton et al., 1988; Jenkinson, 1990), dynamic vegetation distributions (Martín Belda et al., 2022; Weng et al., 2015; Fisher et al., 2015; Liu et al., 2019), radiation-topography interaction (Hao et al., 2021), urban-scale processes (Oleson and Feddema, 2020; Krayenhoff et al., 2020), subsurface multiphase flow (Bisht et al., 2017; Qiu et al., 2024), and land use and management (Huang et al., 2020b; Binsted et al., 2022; Calvin et al., 2019). These improvements not only advance the capability of LSMs to model complex environmental interactions but also facilitate a mechanistic understanding of changes in land-atmosphere interactions under varying environmental conditions. Particularly, they can be used to predict the disturbance of the land surface, for example, Earth's ecosystem and surface hydrology, in response to climate change and to quantify the respective biogeophysical and biogeochemical feedbacks to the climate system (Ban-Weiss et al., 2011; Fisher and Koven, 2020).



Recent advancements in LSMs have broad applications in land-only simulations and within global
climate models (GCMs) to capture the complex interactions surrounding global climate change (Lawrence
et al., 2019; Martín Belda et al., 2022; Wiltshire et al., 2020). However, the application within GCMs does
not allow for the representation of land processes at kilometer scales and extreme events occurring at daily
to weekly scales (such as extreme precipitation and flash drought), which are more relevant to human
society. While regional refinement may appear to be a feasible solution, the associated computational costs
restrict their wide adoption within the weather and climate modeling community. Alternatively, combining
advanced LSMs with Regional Climate Models (RCMs) could facilitate more in-depth examinations of the
climate change impacts on land surfaces and the resulting feedback at scales that have greater relevance to
human society.
The U.S. Department of Energy's Energy Exascale Earth System Model (E3SM) Land Model
(ELM) is an advanced LSM that simulates the exchanges between terrestrial land surfaces and other Earth
system components, enabling us to understand hydrologic cycles, biogeophysics, and the dynamics of
terrestrial ecosystems (Burrows et al., 2020). The Weather Research and Forecasting (WRF) model serves
as an essential tool widely used for regional weather prediction and climate change analysis (Skamarock
and Klemp, 2008). WRF can be run with various LSMs such as Noah, Noah-MP, SSiB, CLM4. It has also
been coupled with CTSM recently (CTSM Development Team, 2024; Ucar, 2020). However, integrating
ELM with WRF enables comprehensive representation of land processes, following recent advancements
in ELM, for more computationally efficient regional modeling applications. For instance, leaf to canopy
upscaling through a two-big-leaf parameterization in ELM enables simulation of the diffuse radiation
fertilization effect (Chakraborty et al., 2022a), and thus better estimates of surface water and carbon budget,
a feature not present in Noah. As another example, ELM incorporates gridwise surface properties such as
leaf area index (LAI), displacement height, and vegetation top and bottom height. In contrast, Noah and its
variants use lookup tables with these properties prescribed for each land cover class, limiting their ability
to capture spatial heterogeneity in surface properties within individual land cover types. Moreover, ELM
simulations at ~km resolution highlight the significance of considering radiation-topography interaction in





simulating surface energy balance and water budget, a process not yet considered by current land models
in WRF (Hao et al., 2021; Yuan et al., 2023).

This study integrates ELM v2.1 with WRF (hereafter named WRF-ELM) using a modified coupler

derived from University Corporation for Atmospheric Research (UCAR)'s Lightweight Infrastructure for
Land-Atmosphere Coupling (LILAC) (Ucar, 2020). We evaluate the model performance using a broad
range of site observations and reanalysis data, providing a benchmark for subsequent model enhancements.
This effort expands the capability of a global LSM, which has been previously used within GCM
frameworks, allowing it to simulate higher resolution land-atmosphere interactions at regional scales. The
introduction and release of WRF-ELM also benefit the ELM user community by providing opportunities
for them to test new land schemes with atmospheric feedbacks and calibrate model parameters in coupled
models.

**2. Methods**
**2.1 Coupler in E3SM**



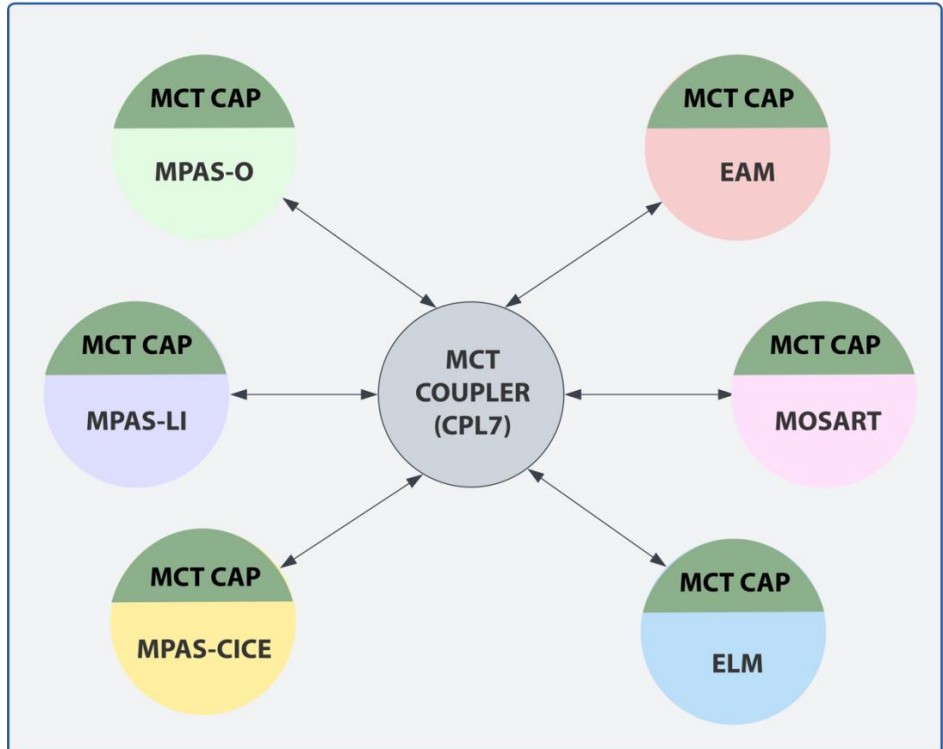

**Figure 1** Schematic diagram of the E3SM model components. The top-level coupler (CPL7) serves as the main program for communication between each component. The Model Coupling Toolkit (MCT) cap in each component provides an interface between CPL7 and the physical core, which is responsible for memory allocation, preprocessing, post-processing, and input and output (I/O).

E3SM adopts a hub-and-spoke architecture to couple the different model components together, as shown in Figure 1. In this architecture, communication between the parallel components is realized via the Model Coupling Toolkit (MCT; (Larson et al., 2005; Jacob et al., 2005)). The top-level coupler, version 7 coupler (CPL7), calls model component initialization, execution, and finalization methods through specified interfaces (Craig et al., 2012). The MCT cap within each component provides an interface between the CPL7 and the physical core, which is responsible for memory allocation, preprocessing, post-processing, and input and output (I/O). Importantly, the inter-component communication is realized only through the



central hub, instead of direct communication with one another. The E3SM coupling framework imposes
strict requirements on how an atmospheric model can communicate with ELM. One particular challenge is
that many atmosphere models – including WRF – expect to run the land model in the middle of the time
step sequence. Accomplishing this in the E3SM architecture can require significant restructuring of the
atmosphere model. For this reason, ELM has not been coupled to atmospheric models in the regional model
community, limiting its ability to address complex scientific challenges at fine resolutions.

**2.2 LILAC-ESMF Coupler**

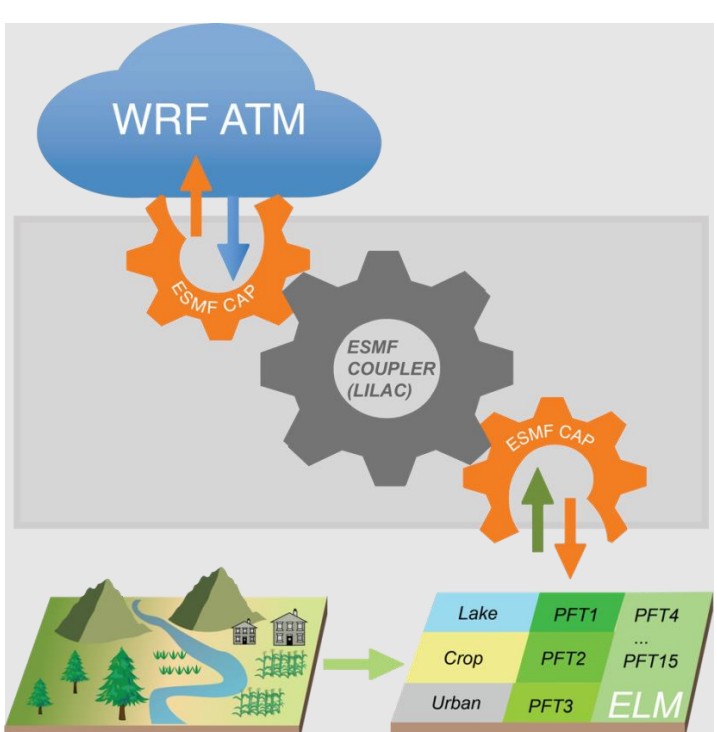


**Figure 2** Schematic diagram of the coupling framework for WRF-ELM. The top-level coupler (LILAC) is
in charge of communication between WRF ATM and ELM. The ESMF Cap within ELM and WRF ATM is
responsible for memory allocation, preprocessing, post-processing, and input and output (I/O).






The traditional way of coupling between LSMs (CLM4, Noah, Noah-MP, and SSiB) and WRF is
through internal subroutines and interfaces within the WRF codebase. This tight coupling means that the
LSM is often compiled and run as an integral part of the WRF model. As the LSMs grow to integrate more
land processes, the tight coupling approach can become less scalable and harder to manage. Additionally,
maintaining the coupled system updated with the latest versions of WRF and LSMs can be challenging due
to the need for synchronized updates and compatibility checks. In contrast, modern approaches such as
LILAC-ESMF offer a more modular and flexible way of coupling, facilitating easier integration and updates
of different model components.
We have developed an ESMF (Hill et al., 2004) Cap which wraps ELM to facilitate seamless
communication with the central hub driver that connects WRF ATM and ELM (Fig. 2). The central hub
driver, LILAC, is developed using ESMF and provides the fundamental functions to support the integration
of an LSM within an RCM, including 1) creating the list of fields passed from WRF ATM to ELM and vice
versa; 2) initializing ESMF Caps for WRF ATM and for ELM); 3) coordinating calls of the ESMF Caps
and ELM and exchanging data between these components; and 4) providing missing atmospheric fields,
specifically for atmospheric aerosols
Within the coupling framework, the ESMF Cap provides the functions of 1) converting the input
data from LILAC to the land model and vice versa; 2) supplying any additional input fields that ELM
requires but are not provided by WRF ATM, for example, gross domestic product, population density, and
lightning that are used to predict fire ignitions in ELM; and 3) setting the domain decomposition and
generating the land mesh. The ESMF cap, which provides the necessary infrastructure to connect LILAC
and ELM physics, serve as an example for similar coupling work between other LSMs and RCMs.

**2.3 Exchange variables between WRF and ELM**
ELM is driven by meteorological forcings including precipitation, downward shortwave radiation,
downward longwave radiation, zonal wind at reference height ($z_{atm}$), meridional wind at $z_{atm}$, pressure at
$z_{atm}$, specific humidity at $z_{atm}$, and air temperature at $z_{atm}$. In the coupled version, the meteorological forcings



are provided by WRF ATM with the ELM model timestep set to match the integration timestep in the WRF
ATM. The reference height refers to the height of the lowest atmosphere model level. The radiation scheme
in WRF further splits the shortwave radiation to direct and diffuse components, as well as visible and near-
infrared radiation. Precipitation is divided into rainfall and snowfall based on the frozen precipitation ratio,
which are then inputted into the ELM. The ELM output includes skin temperature, 2-m air temperature, 2-
m specific humidity at the surface, friction velocity, surface albedo, sensible heat flux, latent heat flux,
ground heat flux, surface emissivity, and roughness length for momentum and heat transfer, which will be
exchanged with the WRF ATM component.

**2.4 Mesh data and surface parameters**
In addition, mesh data is used in the WRF ATM to define the latitude and longitude of the grid. The
domain information is necessary for the coupler and the land model during runtime. These data include a
mask that informs the land model where to run and a land fraction that the coupler uses to combine fluxes
from various surface types over a grid cell. The surface data configures the spatially implicit features (e.g.,
spatial fraction coverage, leaf and soil albedo, leaf and soil emissivity, etc.) of subgrid elements within grid
cells (topographic unit, land cover, soil columns, and vegetation).
While a regular latitude/longitude grid is widely used for domain and surface data in the land-only
mode, when coupled with WRF ATM, ELM needs to adopt the Lambert Conformal projection used in WRF.
To create a domain file of Lambert Conformal projection, a grid descriptor file based on the WRF Pre-
Processing System (WPS) output (e.g., geo_em.d01) needs to be created, which is then used to create the
domain file used in ELM. A similar workflow is needed for surface data, which contains a large number of
input files that need to be interpolated by the land model. To generate both domain files and surface data, we
employ the ELM preprocessing tools that derive the input data and grid descriptor files for each dataset,
produce mapping files from the input data grid to our target grid, and then use the mapping weight files for
interpolation.



**2.5 Parallelization**

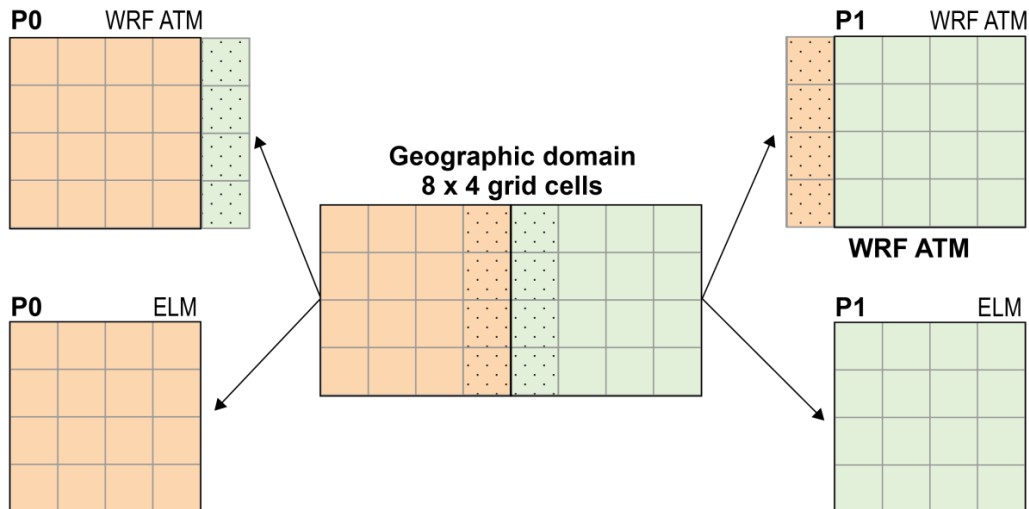


**Figure 3** Schematic of parallel domain decomposition scheme in WRF-ELM. The dotted area indicates
'halo' arrays in which memory is shared between processors (P0 and P1). WRF ATM and ELM are
calculated under the same processor.

200   Instead of adopting ELM's native round-robin domain decomposition strategy, our parallelization

strategy for WRF-ELM is to use geographic domain decomposition, as in WRF ATM. As shown in Fig. 3,
different grid cells in the model's physical domain are running on separate processors pre-assigned by the
user. On each processor, ELM within WRF employs parallel I/O to read atmospheric forcings, uses the
surface properties and land-use datasets to configure individual land cells, and then conducts massively
parallel simulations over these grid cells within each subdomain independently. In WRF ATM, the 'halo'
arrays share memory between processors, and message passing between processors is accomplished using
the message passing interface (MPI; (Gropp et al., 1996)).

**3. Model Validation**



**3.1 WRF-ELM configuration**


For our first WRF-ELM application, we study the land-atmosphere interactions over the Great
Lakes Region (GLR), a hydrodynamically complex and heavily populated region with both natural surface
heterogeneity and significant land management practices. This domain also includes the world's largest
freshwater system, comprising of Superior, Michigan, Huron, Erie, and Ontario Lakes. This region is the
focus of the U.S. Department of Energy's (DOE's) Coastal Observations, Mechanisms, and Predictions
Across Systems and Scales, Great Lakes Modeling (COMPASS-GLM) project, which has an overall goal
of developing a fully coupled (lake-land-atmosphere) regional earth system model centered on the GLR
(Kayastha et al., 2023). Here, we report the initial implementation of the WRF-ELM framework to support
its ability to capture atmospheric, coastal, urban, and rural interactions, providing a baseline reference
solution for further model development.

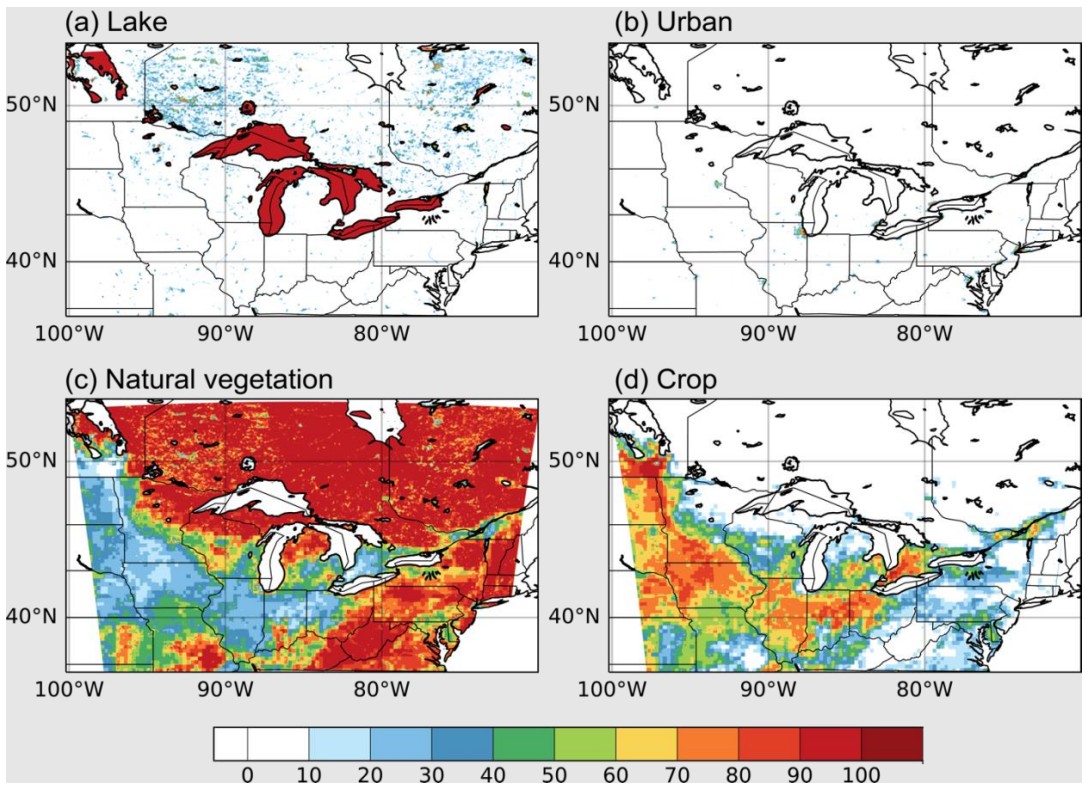




**Figure 4** Fractional coverage (%) of major land unit (a) lake, (b) urban, (c) natural vegetation, and (d) crop
used in the WRF-ELM.

The RCM used in the numerical simulation is based on the WRF model version 4.4.2 with the

Advanced Research WRF dynamic core (Skamarock and Klemp, 2008). Following Wang et al. (2022a), the
model domain is centered at 45.5°N and 85.0°W and has dimensions of 544 × 485 grid points in the west–
east and south–north directions. The simulation domain covers the GLR, with a spatial resolution of 4 km
(Fig. 4). Fifty vertical layers from the surface to 50 hPa are adopted with denser layers at lower altitudes to
sufficiently resolve the PBL. We conduct 5 ensemble members in 2018, starting with initial conditions 12
hr apart between 0000 UTC on 12 May and 0000 UTC on 14 May and ending on 0000 UTC 1 September
2018. The resulting simulations are analyzed during June, July, and August (JJA) 2018.

**Table 1** Model Configuration in WRF and ELM.

| WRF specific options and schemes | |
| --- | --- |
| Meteorological IC/LBCs | ERA5 |
| Microphysics | Thompson microphysics |
| Radiation | RRTMG for longwave and shortwave |
| Land surface | ELM or CTSM |
| Planetary boundary layer | YSU scheme |
| Lake surface temperature | NOAA GLSEA |
| ELM/CTSM input data | |
| Land use and land cover | ELM/CTSM default parameter |
| Vegetation | ELM/CTSM default parameter |
| Soil color | ELM/CTSM default parameter |
| topography | ELM/CTSM default parameter |
| Number of plant functional types (PFT) | 16 |


The meteorological initial condition (IC) and lateral boundary conditions (LBCs) have been derived

from the ECMWF Reanalysis v5 (ERA5; (Hersbach et al., 2020)) at 0.25° horizontal resolution and 3-hour
temporal intervals (Table 1). The WRF model incorporates the Thompson microphysics (Thompson et al.,
2004; Thompson et al., 2008), the Rapid Radiative Transfer Model for GCMs longwave and shortwave
schemes (Iacono et al., 2008), and the Yonsei University planetary boundary layer scheme (Hong and Lim,



2006). We turn off cumulus parameterization, considering the convection-permitting resolution of the
ensemble simulations. The lake skin temperature is obtained from NOAA Great Lakes Surface
Environmental Analysis (GLSEA) data set (Schwab et al., 1992) derived from Advanced Very High-
Resolution Radiometer.

For the land surface model, we adopt ELM with satellite phenology (ELM-SP) mode which utilizes

seasonal varying leaf area index prescribed based on the MODIS data. The default ELM land surface
parameters have been used in the coupled model simulation, including land use and land cover information,
vegetation biogeophysical properties, soil properties, and topography. The surface parameter is also
applicable in CTSM (Table 1). A detailed description of ELM/CTSM default parameter can be found in (Li
et al., 2024). The current version of WRF-ELM does not enable biogeochemistry (ELM-BGC) mode and
thus does not simulate carbon and nitrogen cycles. In addition, we also conduct simulations using the WRF
coupled with Community Terrestrial Systems Model (CTSM ctsm5.1.dev114) (Lawrence et al., 2019)
(WRF-CTSM hereafter), which can be used to compared with WRF-ELM's performance in capturing the
land-atmosphere exchanges of energy and water fluxes. CTSM is also referred to the community land model
version 5 (CLM5) afterwards.



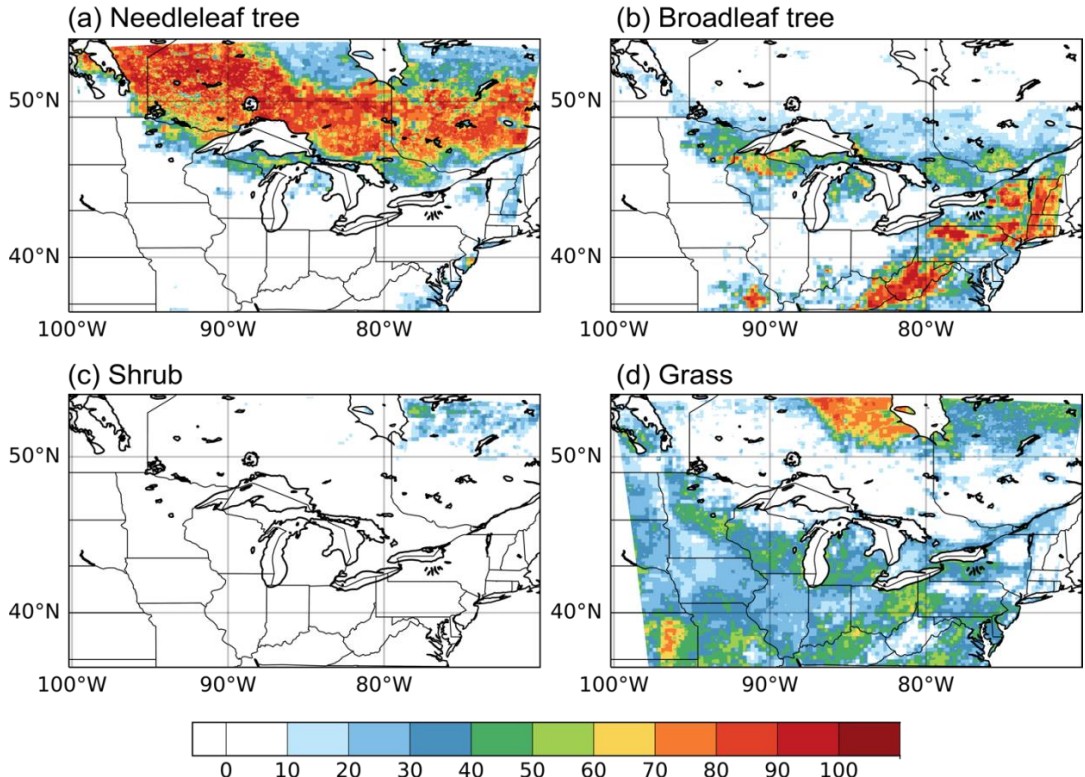

**Figure 5** Fractional coverage (%) of major plant functional types (a) needleleaf forest (deciduous and evergreen combined), (b) broadleaf forest (deciduous and evergreen combined), (c) shrub, and (d) grass used in the WRF-ELM.

It is noteworthy that there are several distinctions between WRF-ELM and the version of WRF-CTSM we use here. WRF-CTSM aims for a relatively fast calculation speed, thus it has simplified the description of land cover and kept the single dominant land unit and single dominant PFT. In our simulation region, WRF-CTSM identifies the Great Lakes in the center of the simulation domain, with the natural vegetation prevailing in the northern and southeastern regions, and crops dominating the southwestern areas (Fig. 4). On the other hand, WRF-ELM preserves the comprehensive description of subgrid heterogeneity. As a result, the fluxes calculated from various surface types are merged using a weighted-average method before transferring to the upper-level WRF ATM. This is particularly important in regions with mixed



vegetation types, such as the southwestern part of our study domain. Moreover, within the natural
vegetation land unit, WRF-ELM simulates the blend of needleleaf and broadleaf trees (evergreen and
deciduous combined) around the Great Lakes and the mixture of crops and grasses in the southwestern part
of the domain (Fig. 5).

**3.2 Data for validation**

**Table 2** Dataset for validation in the study.

| Dataset | Variables | Spatial resolution | Temporal resolution | Reference |
|---------|-----------|--------------------|--------------------|-----------|
| ASOS | Air temperature at 2-m, Dew point | point | Hourly | (Nadolski, 1992) |
| AmeriFlux | Latent heat, Sensible heat | point | Hourly | (Law, 2005) |
| Daymet | Maximum air temperature at 2-m, Maximum air temperature at 2-m, Precipitation | 1 km | Monthly | (Thornton et al., 2022) |
| NLDAS | Air temperature at 2-m, Precipitation | 0.125 ° | Monthly | (Xia et al., 2012) |
| ERA5-Land | Air temperature at 2-m, Precipitation, Latent heat, Sensible heat | 9 km | Monthly | (Muñoz-Sabater et al., 2021) |


Observational and reanalysis data from multiple sources have been used to evaluate WRF
simulation results (Table 2). We select 12 paired sites from the Automated Surface Observing System
(ASOS) to acquire 5-minute 2-meter air temperature (Ta) and 2-meter dew point temperature over the urban
and rural area in the GLR (https://www.ncei.noaa.gov; last accessed: November 2023). The 2-meter relative
humidity (RH) is derived from Ta and dew point. We compute hourly averages of Ta and RH from the 5-
minute data to match the hourly WRF outputs.



**Table 3** AmeriFlux site information (LCF: land cover type; DBF: deciduous broadleaf tree; MF: mixed
forest; NEON: National Ecological Observatory Network)

| Site ID | Latitude | Longitude | LCF | PI(s) | DOI |
|---------|----------|-----------|-----|-------|-----|
| US-xST | 45.5089 | -89.5864 | DBF | NEON | https://doi.org/10.17190/AMF/1617737 |
| US-xTR | 45.4937 | -89.5857 | DBF | NEON | https://doi.org/10.17190/AMF/1634886 |
| US-WCr | 45.8059 | -90.0799 | DBF | Ankur Desai | https://doi.org/10.17190/AMF/1246111 |
| US-xUN | 46.2339 | -89.5373 | MF | NEON | https://doi.org/10.17190/AMF/1617741 |
| US-PFa | 45.9459 | -90.2723 | MF | Ankur Desai | https://doi.org/10.17190/AMF/1246090 |
| US-Syv | 46.242 | -89.3477 | MF | Ankur Desai | https://doi.org/10.17190/AMF/1246106 |


In addition, we collect measurements of latent heat (LH) and sensible heat (SH) from six flux tower
sites provided by AmeriFlux (http://ameriflux.lbl.gov; last accessed: November 2023). Initially, 16
AmeriFlux sites have been selected within our study domain for the JJA 2018 period, which included
measurements over grassland, mixed forest, and deciduous broadleaf forest. However, ten sites are filtered
out because their land cover types differ from the dominant ones used in WRF-CTSM. The latitudes and
longitudes of selected sites have been documented in Table 3. The hourly LH and SH data from AmeriFlux
have been reduced to daily averages to validate the model simulation of surface energy fluxes.
We also acquire reanalysis datasets to evaluate the model performance in simulating the climate
variables and energy fluxes. All datasets are resampled using bilinear interpolation to a 4 km resolution to
align with the WRF grids. We employ the Daymet dataset from https://daymet.ornl.gov (last accessed:
October 2023), which provides daily, gridded (1 km × 1 km) estimates of solar radiation, 2-meter maximum
(Tmax) and minimum (Tmin) temperature, precipitation (PRE), snow water equivalent, and water vapor
across the CONUS (Thornton et al., 2022). It uses local regression algorithms to interpolate and extrapolate
daily meteorological observations from Global Historical Climatology Network (GHCN). Daymet
considers the effects of elevation on climate and generates daily meteorological variables for a particular
grid cell using the weighted linear regression-based approach. We download monthly Tmax, Tmin, and



precipitation from Daymet version 4.5, and average the temperatures to compare against model simulated
daily mean Ta.
Monthly Ta from the North American Land Data Assimilation System version 2 (NLDAS) with
Noah LSM is used as an additional source of reanalysis data to evaluate WRF-ELM. These data are
available beginning in 1979 at a 0.125° resolution (Xia et al., 2012). NLDAS constructed a forcing dataset
from a daily gauge-based precipitation analysis, bias-corrected shortwave radiation, and surface
meteorology reanalyses from North American Regional Reanalysis (NARR) to drive four different LSMs
to derive surface fluxes and state variables. We acquire the product derived using the Noah model
(https://disc.gsfc.nasa.gov; last accessed: October 2023) because it is one of the most commonly used LSMs
and has been frequently coupled with climate and atmospheric models.
The ERA5-Land reanalysis provides surface variables at the 0.1° x 0.1° resolution (Muñoz-Sabater,
2019). The data are produced under the offline mode forced by meteorological fields from ERA5 (Muñoz-
Sabater et al., 2021), without coupling to the atmospheric module of the ECMWF's Integrated Forecasting
System. ERA5-Land datasets have also been widely used for a variety of land condition assessments (Pelosi
et al., 2020; Stefanidis et al., 2021; Wang et al., 2022b). We acquire monthly Ta, SH, and LH in ERA5-
Land from Google Earth Engine (collection ECMWF/ERA5_LAND/MONTHLY_AGGR; last accessed:
October 2023).
Lastly, we acquire precipitation data from the National Centers for Environmental Prediction
(NCEP) Stage IV dataset (Lin and Mitchell, 2005), a gridded product with 4 km spatial and hourly temporal
resolution that covers the period from 2002 to the present. NCEP compiles the Stage IV product using data
from 140 radars and approximately 5,500 gauges across the CONUS. Stage IV provides highly accurate
precipitation estimates, particularly for medium to heavy precipitation, and has therefore been widely used
as a reference for precipitation evaluation (Nelson et al., 2016).

**3.3 Results**
**3.3.1 Temperature**



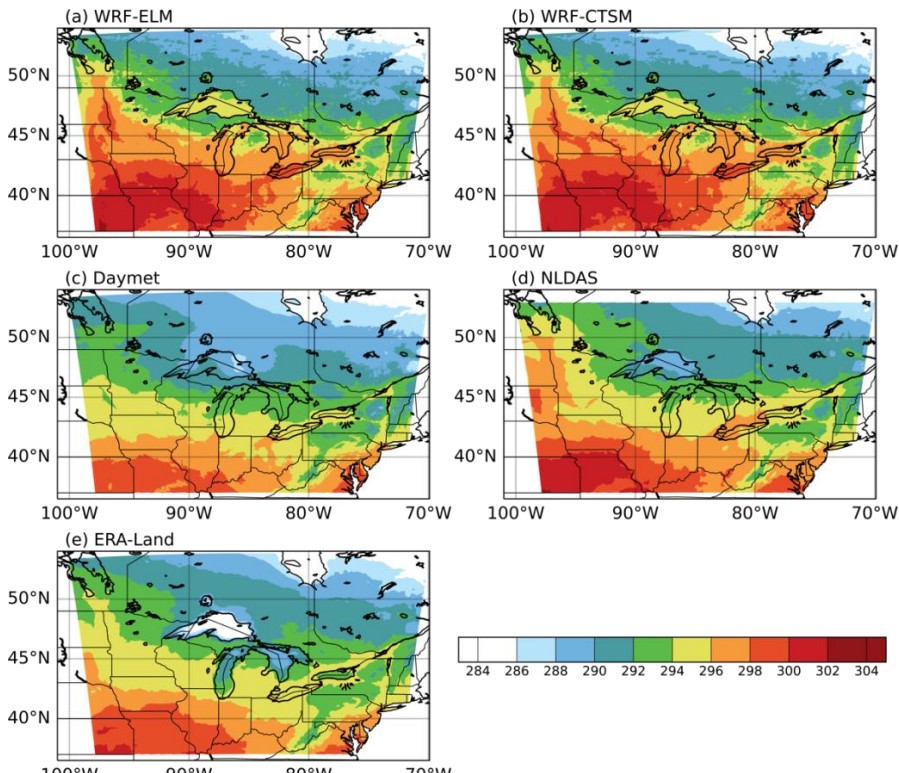


**Figure 6** June-July-August mean 2-m air temperature (K) in (a) WRF-ELM, (b) WRF-CTSM, (c) Daymet,

(d) NLDAS, and (e) ERA-Land. The numbers on the top right of (c)-(f) indicate the spatial correlation

coefficient between each reanalysis product and the two simulation results.


**Table 4** Evaluation metrics of June-July-August 2-m air temperature between each model result and the

reanalysis product. CORR: spatial correlation coefficient; RMSE: Root mean square error.

|          |      | Daymet | NLDAS | ERA-Land |
|----------|------|--------|-------|----------|
|          | Bias | 1.70   | 0.34  | 1.20     |
| WRF-ELM  | CORR | 0.94   | 0.94  | 0.86     |
|          | RMSE | 2.18   | 1.43  | 2.30     |
|          | Bias | 1.79   | 0.43  | 1.29     |
| WRF-CTSM | CORR | 0.94   | 0.93  | 0.86     |
|          | RMSE | 2.30   | 1.57  | 2.40     |


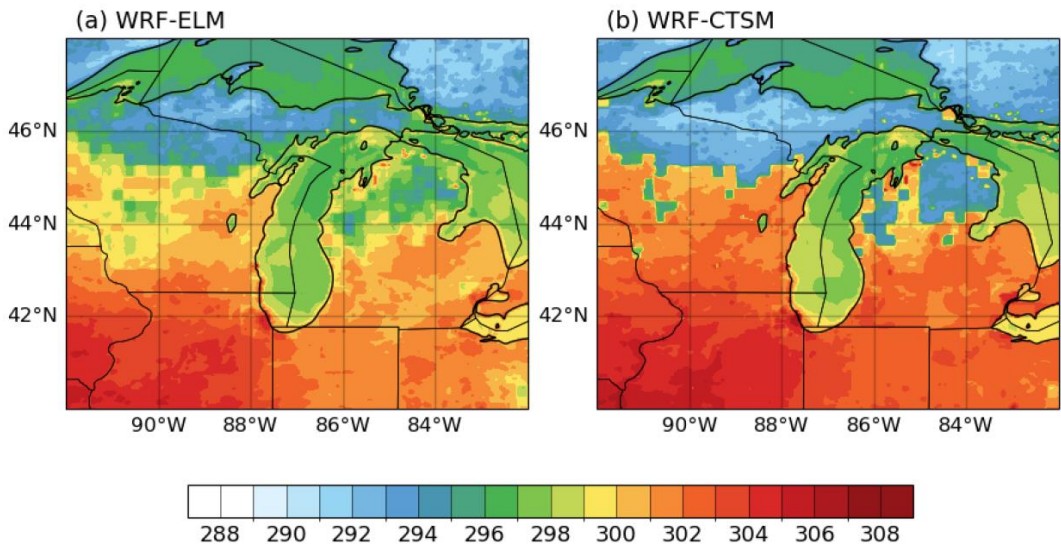

**Figure 7** June-July-August mean skin temperature (K) in (a) WRF-ELM, (b) WRF-CTSM, zoomed-in view

focuses on the area surrounding Lake Michigan

The spatial distribution of Ta from the WRF-ELM and WRF-CTSM models, along with reanalysis

data such as Daymet, NLDAS, and ERA5-Land, is illustrated in Figure 6. Both WRF-ELM and WRF-

CTSM have reasonably captured the spatial pattern observed in the reanalysis datasets, demonstrating a

spatial correlation coefficient (CORR) ranging from 0.86 to 0.95 (Table 4). The highest CORR is observed

with Daymet, while the lowest one is with ERA5-Land. Both models exhibit a warm bias compared to

reanalysis products. However, WRF-ELM shows a slightly lower bias and RMSE compared with WRF-

CTSM (Table 4). Additionally, WRF-ELM displays a smoother gradient in comparison to WRF-CTSM,

particularly over the GLR where needleleaf trees, broadleaf trees, grasses, and croplands coexist (Fig. 7).



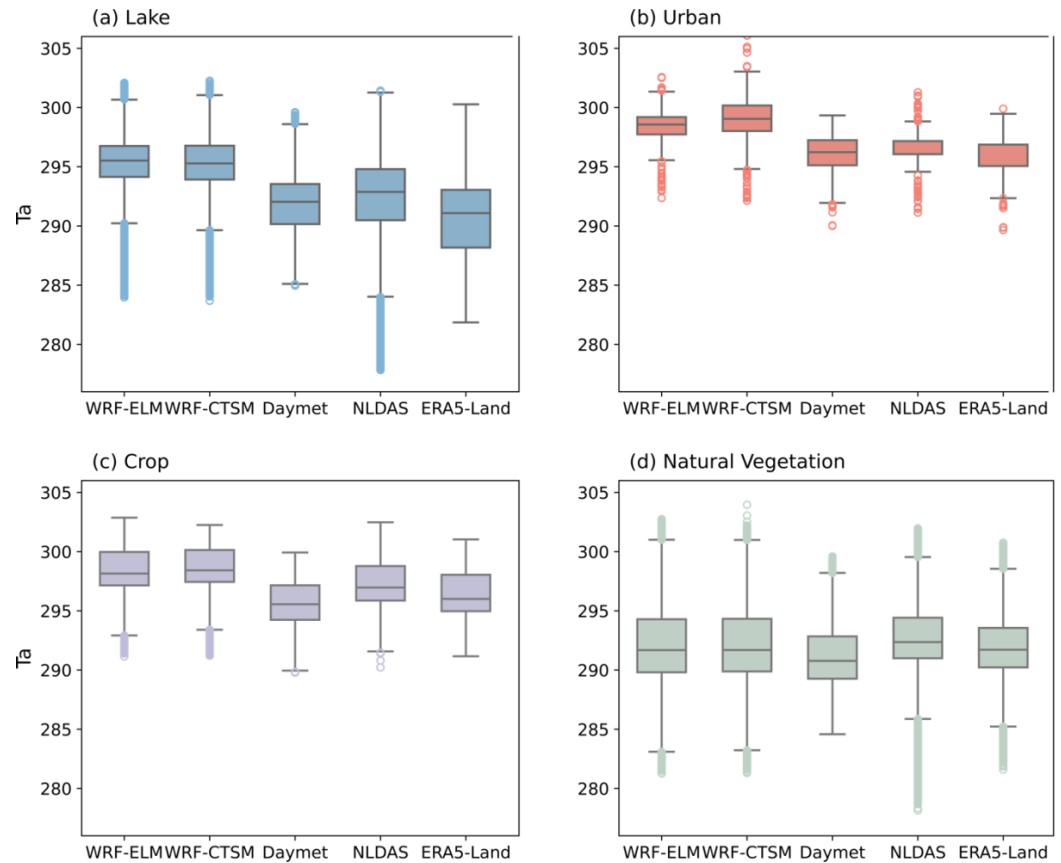

**Figure 8** Boxplots of June-July-August 2-m air temperature (K) over (a) lake, (b) urban, (c) crop, and (d) natural vegetation in simulations and reanalysis products.

**Table 5** June-July-August 2-m air temperature over each land unit in simulations and reanalyses.

|  | WRF-ELM | WRF-CTSM | Daymet | NLDAS | ERA5-Land |
|---|---|---|---|---|---|
| Lake | 295.5 | 295.4 | 292.1 | 292.3 | 290.6 |
| Urban | 298.5 | 299.0 | 296.2 | 296.7 | 296.0 |
| Crop | 298.4 | 298.6 | 295.8 | 297.4 | 296.5 |
| Natural Vegetation | 292.6 | 292.6 | 291.7 | 292.9 | 292.4 |




Despite the overall good performance of model simulation of Ta, it is slightly different among

different land units (Fig. 8). The largest warm bias is found over the lake surface, in which both models
have overestimated Ta by 3-5 K (Table 5, Fig. 8). For urban and crop areas, the WRF-ELM and WRF-
CTSM show a slightly warmer temperature by 2-3 K than all reanalysis data, which makes sense since
reanalysis datasets do not capture urban-scale warming signals (Chen et al., 2024). The Ta over the natural
vegetation is well captured, with the average value in both models within the range of average Ta over all
datasets.

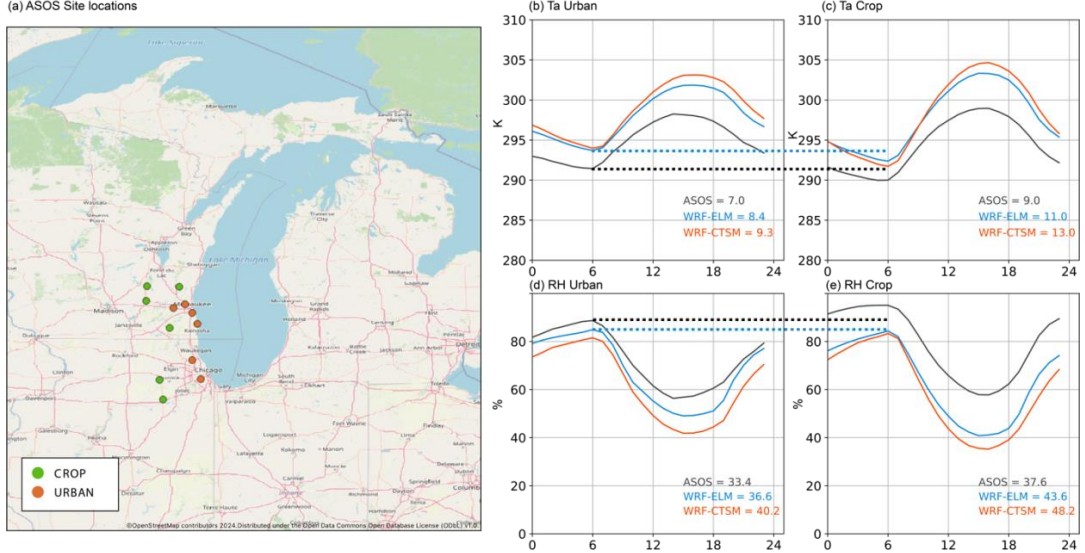


**Figure 9** (a) The location of ASOS sites. (b-c) June-July-August averaged hourly 2-meter air temperature
over (b) urban and (c) crop land units for ASOS, WRF-ELM, and WRF-CTSM. (d-e) The same as (b-c) but
for 2-meter relative humidity. The numbers in (b-e) indicate the diurnal ranges of air temperature and
relative humidity from ASOS, WRF-ELM, and WRF-CTSM. The dash lines highlight the nighttime Ta and
RH when urban and crop contrasts are significant.

We use ASOS sites to examine the representation of urban heat island (UHI; (Rizwan et al., 2008))

and urban dry island (UDI; (Chakraborty et al., 2022b)) effects in WRF-ELM and WRF-CTSM. Six urban





sites on the west coast of Lake Michigan have been selected, and correspondingly, six crop sites near the
urban sites are chosen as pair sites (Fig. 9a). Compared to the adjacent rural sites, the urban sites exhibit a
higher minimum Ta during the night and early morning, leading to a reduced diurnal temperature range of
7.0 K, compared to the 9.0 K range over the crop sites (Figs. 9b-c). During the late morning to noon, the
lake breeze tends to cool the urban air temperature, resulting in lower daily maximum Ta than over the crop
areas (Wang et al., 2023). In the afternoon, the urban sites display a more gradual temperature change slope
than the rural sites, attributable to the cumulative heating effect of solar radiation absorption and heat release
by urban materials throughout the day (Soltani and Sharifi, 2017). The UDI effect is also discernible in the
2m RH in ASOS observations, with urban areas exhibiting lower values at night (Figs. 9d-e). Both WRF-
ELM and WRF-CTSM have captured the warmer Ta and lower RH during the night and the smaller diurnal
range of Ta and RH in urban compared with crop sites. Notably, WRF-ELM generally demonstrates smaller
biases in both Ta and RH than WRF-CTSM (Figs. 9).

**3.3.2 Energy fluxes**



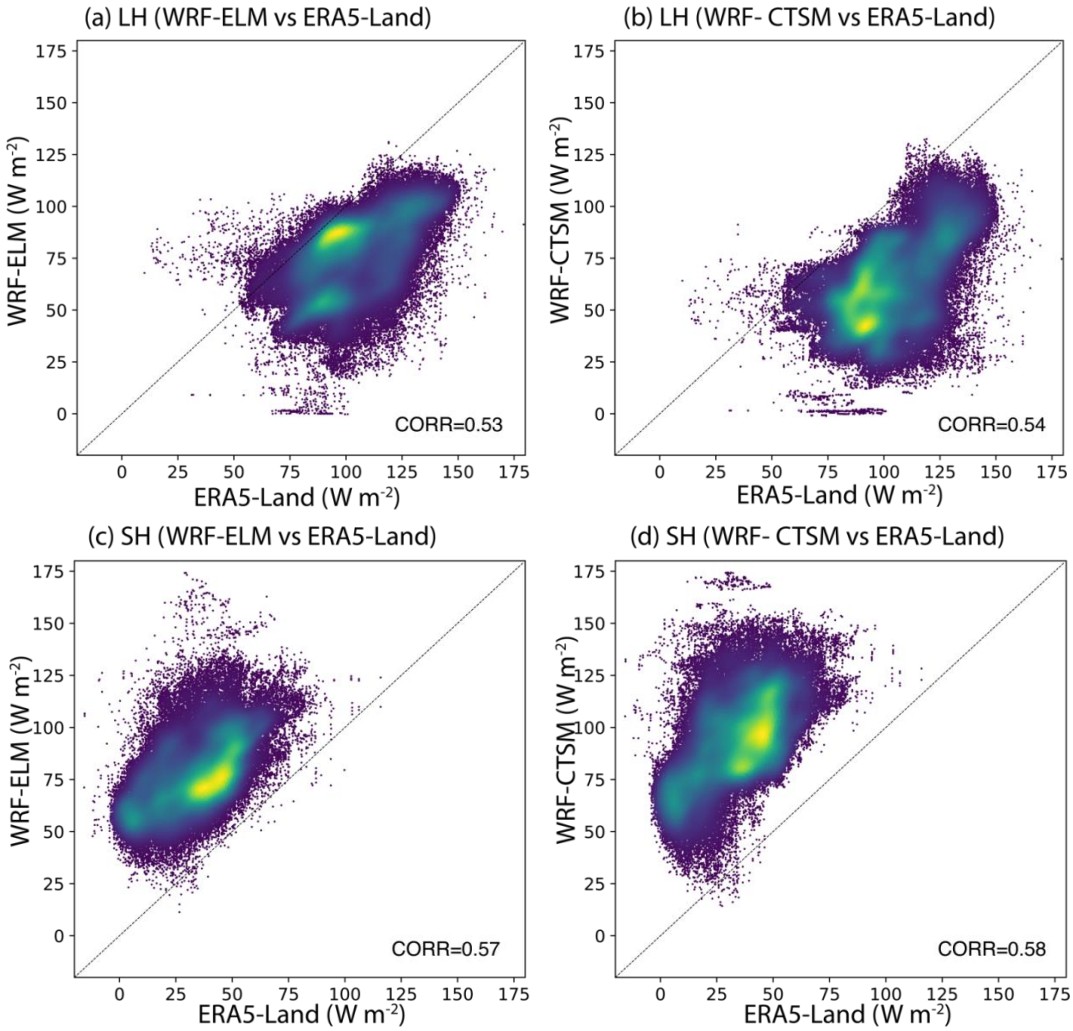

**Figure 10** Comparison of latent heat over natural vegetation land unit between (a) WRF-ELM and ERA5-Land and (b) WRF-CTSM and ERA5-Land. (c)-(d) Same as (a)-(b) but for sensible heat. Each point represents the JJA mean surface fluxes in a grid.



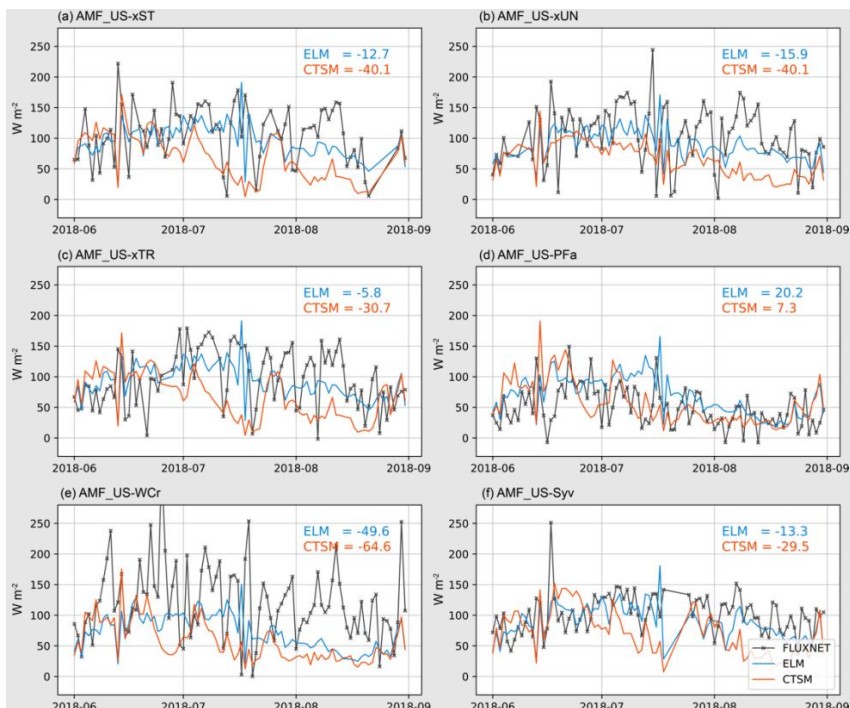


**Figure 11** June-July-August averaged daily LH fluxes from six AmeriFlux sites and the corresponding
model grids. The numbers indicate biases between WRF-ELM (or WRF-CTSM) and AmeriFlux.

We evaluate the simulated LH and SH fluxes from model simulations against ERA5-Land. The

spatial CORR ranging from 0.53 to 0.58 (Fig. 10). An underestimation of LH and an overestimation of SH
are evident for both WRF-ELM and WRF-CTSM compared to ERA5-Land. A further comparison of daily
LH values from six AmeriFlux sites over deciduous broadleaf forest is illustrated in Figure 11. The observed
temporal variations of LH, largely influenced by incoming solar radiation and precipitation, are roughly
captured in both WRF-ELM and WRF-CTSM. Compared to the daily observations from AmeriFlux sites,
WRF-ELM demonstrates a superior ability to reproduce the observed magnitude of LH and exhibits a
smaller bias than WRF-CTSM (Fig. 11)

**3.3.3 Precipitation**




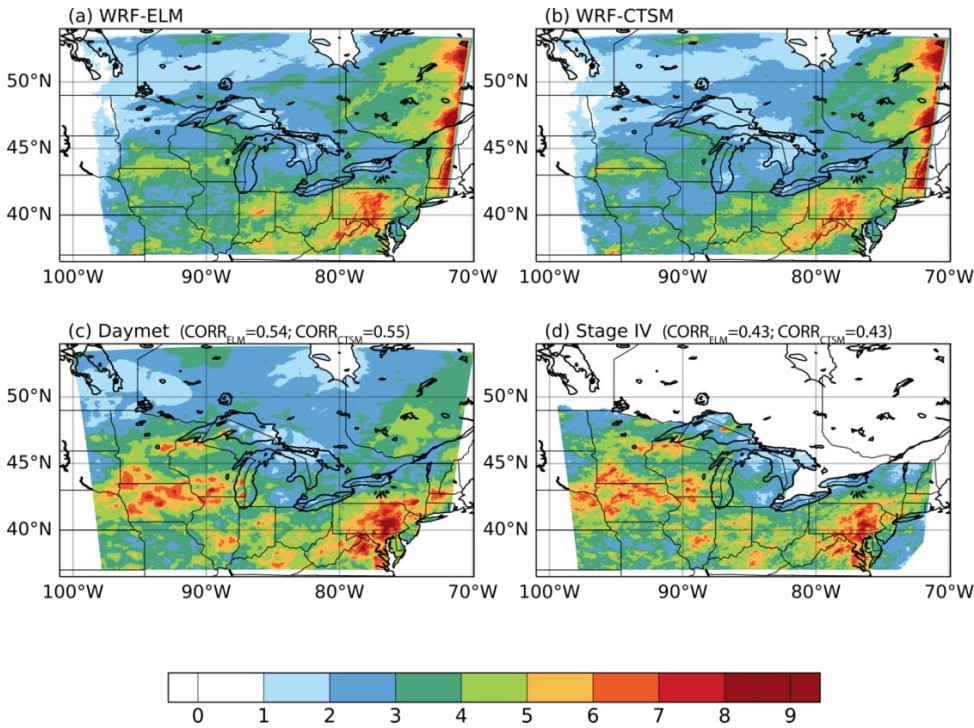


**Figure 12** The spatial distribution of June-July-August precipitation (mm d$^{-1}$) in (a) WRF-ELM, (b) WRF-

CTSM, (c) Daymet, and (d) ST4. The numbers on the top right of (c)-(d) indicate the CORR between each

observational product and the two simulation results.


Figure 12 presents the spatial distribution of precipitation from models and observations. It is

important to note that Stage IV primarily focuses on the CONUS region, while significant areas of our

simulation domain in Canada remain uncovered. Compared with the Daymet (PRE$_{Daymet}$ = 3.55 mm d$^{-1}$),

both WRF-ELM and WRF-CTSM capture the regional mean value (PRE$_{WRF-ELM}$ = 3.14 mm d$^{-1}$ and PRE$_{WRF-}$

$_{CTSM}$= 2.96 mm d$^{-1}$) and the spatial distribution of precipitation, exhibiting CORR ranging from 0.43 to 0.55.

The precipitation over the southeastern part of our study domain is well captured while that on the western

side of Lake Michigan is slightly underestimated, with WRF-ELM demonstrating a lower bias than WRF-

CTSM. This underestimation of precipitation aligns with the underestimation of latent heat and



evapotranspiration, suggesting that suppressed evapotranspiration may reduce moisture availability and
transport, particularly to the western GLR. Conversely, an overestimation of precipitation is evident along
the eastern boundary of our study domain.

**4. Discussion and Conclusions**
This study introduces a framework integrating the state-of-the-art land surface model, ELM, with
the widely used regional weather and climate model, WRF, named WRF-ELM. Moving beyond the
traditional way of coupling between LSMs and WRF through internal subroutines within the WRF codebase.
We adopt the LILAC-ESMF framework, a modular approach which maintains the integrity of the ELM's
source code structure and facilitates the transfer of future developments in ELM to WRF-ELM. After
coupling the two models, simulations using WRF-ELM have been conducted over the Great Lakes Region,
and their performance has been evaluated against observations and reanalysis data from multiple sources
and the WRF-CTSM simulations. These model simulations have been conducted at a resolution of 4 km ×
4 km, facilitating direct model validation and verification with various data sources. The use of seasonal
mean simulation outputs and diurnal cycles showcases the capabilities of WRF-ELM in representing the
temporal and spatial variations of water and energy cycles over the Great Lakes Region.
In general, our findings suggest that the newly coupled WRF-ELM effectively captures the spatial
distribution of surface state variables and fluxes across the GLR. The model displays a smoother gradient
in surface skin temperature than WRF-CTSM, due to the representation of sub-grid features within grid
cells. The model's performance is particularly reasonable over the natural vegetation, while a minor warm
bias is detected over crop and urban grids.
The slight overestimation of air temperature in crop regions could potentially be mitigated by
incorporating a more realistic representation of crops, such as crop rotation and irrigation. Additionally, the
application of spatially varying crop parameters closely captures the observed magnitude and seasonality
of carbon and energy fluxes compared to the observations (Sinha et al., 2023). However, these
improvements have only been tested using the land-only ELM. Our generalized coupling framework



supports future studies of sophisticated crop-atmosphere interactions at finer spatial resolution than those
achieved with coarse GCM simulations.

In addition, the UHI effects in cities surrounding the GLR are generally captured in both WRF-

ELM and WRF-CTSM, as indicated by the warmer night temperature in the cities. While there is an
overestimation of UHI compared to ASOS, this could be due to the simplified urban representation in ELM.
For instance, the urban surface emissivity in CLM, and thus ELM due to the shared model structure, is
reported to be noticeably lower than the values derived from satellites, resulting in a surface UHI effect that
is significantly higher than satellite-derived values (Chakraborty et al., 2021). Another potential
contributing factor could be the lack of representation of urban vegetation. The presence of vegetation tends
to mitigate the UHI effect (Paschalis et al., 2021) , and its absence in the urban subgrid would lead to an
overestimation of UHI values, all else remaining equal.

Our research develops the WRF-ELM framework and provides the first assessment of its

capabilities through high-resolution model simulations that fully capture expected patterns of land-
atmosphere interactions. Based on the validation and assessment of WRF-ELM results, this study delivers
a baseline reference, identifies common model biases in high-resolution regional applications, and proposes
pathways for subsequent model development for ELM, as well as the coupled model. The coupled model
provides an opportunity to investigate the impact of more sophisticated land processes, such as plant
hydraulics, dynamic vegetation distributions, and soil biogeochemistry, on weather and climate predictions.

**Author contributions:** HH designed the study, implemented the parameterization, performed the
simulations, analyzed the results, and drafted the original paper. YQ designed the study, discussed the results,
and edited the paper. GB, TT, BS, YL, and WS helped with the coupling design. JW, TT, DH, JL, ZY, PX,
EC and RH discussed the results and edited the paper.

**Code Availability:** The description and codes of E3SM v2.1 (including ELM v2.1) are publicly available
at        https://doi.org/10.11578/E3SM/dc.20230110.5        and        https://github.com/E3SM-



Project/E3SM/releases/tag/v2.1.0 (last access: 12 May 2023), respectively. Starting from ELM 2.1, the
model codes for WRF-ELM coupling described in this paper are available at
https://github.com/hhllbao93/ELM and https://doi.org/10.5281/zenodo.11289807 (Huang, 2024).

**Competing interests:** The authors declare that they have no conflict of interest.

**Acknowledgement**: The authors acknowledge the CTSM developer teams for making the LILAC release
available including Mariana Vertenstein, Negin Sobhani, Samuel Levis, David Lawrence, Michael Barlage,
Joe Hammann, and Erik Kluzek.

**Financial support:** This study is supported by COMPASS-GLM, a multi-institutional project supported
by the U.S. Department of Energy (DOE), Office of Science, Office of Biological and Environmental
Research, Earth and Environmental Systems Modeling program. T.C.'s contribution was also supported by
the DOE, Office of Science, Biological and Environmental Research program through an Early Career
award. The Pacific Northwest National Laboratory is operated for DOE by Battelle Memorial Institute
under contract DE-AC05-76RL01830.




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
