# Peer review of "WRF-ELM v1.0: a Regional Climate Model to Study Atmosphere-Land Interactions Over 2 Heterogeneous Land Use Regions 3 Huilin Huang1\*, Yun Qian1\*, Gautam Bisht1, Jiali Wang2, Tirthankar Chakraborty1, Dalei Hao1, Jianfeng Li"

_EGUsphere, 2024_

## Author Comment (AC1)

We sincerely appreciate the referees for their valuable and insightful comments on our manuscript. The feedback is instrumental in enhancing the quality and clarity of our research. These comments are not only valuable but also serve as a critical resource for improving various aspects of our article, including methodology, data interpretation, and overall presentation. We have taken each comment seriously and conducted a thorough review of our manuscript to ensure that we comprehensively address all concerns raised by the referees.

This response document provides a detailed account of the changes implemented in relation to each specific comment from the referee. For ease of reference, referee comments are presented in black, while authors' responses are provided in dark blue. The revised manuscript is highlighted in blue. The line numbers correspond to the revised version of the manuscript (track changed).

Reviewer #1

This paper integrates ELM v2.1 with the Weather Research and Forecasting (WRF) Model through a modified Lightweight Infrastructure for Land Atmosphere Coupling (LILAC) framework, enabling affordable high-resolution regional modeling by leveraging ELM's innovative features alongside WRF's diverse atmospheric parameterization options. High-resolution (4 km) WRF-ELM ensemble simulations over the Great Lakes Region (GLR) in the summer of 2018 are evaluated with observations, reanalysis data, and the WRF-CTSM. The manuscript is very well-written and has a very nice flow to it. I have some minor comments and suggestions to strengthen the manuscript:

1. Figure 6: The numbers on the top right of (c)-(f) indicate the spatial correlation coefficient between each reanalysis product and the two simulation results. However, the numbers and figure 6f are missing.

R: Thank you for pointing this out. We include fig (f) in caption by mistake. It has been removed in the text.

2. The partitioning of surface energy between latent and sensible heat fluxes plays an important role in regulating heat and water exchange between the land surface and the atmosphere. The spatial distributions of latent and sensible heat fluxes should be evaluated in the manuscript.

R: We agree with the reviewer that the evaluation of spatial distributions of latent and sensible heat fluxes would be important. We have now included 1) spatial distributions of sensible and latent heat and 2) scatter plot of evaporative fraction (LH/(LH+SH)) to explicitly evaluate the partitioning between latent and sensible heat in the updated Fig. 10. For most grids, we found both WRF models systematically underestimate LH while overestimate SH, producing a lower evaporative fraction compared to the observational data. We have included a comprehensive discussion regarding the spatial distribution of SH and LH, and the scatter plot of the evaporative fraction in the revised manuscript (Lines 403-414).

"We evaluated the simulated LH and SH fluxes from the WRF model simulations against ERA5-Land reanalysis data. The spatial correlation coefficients (CORR) range from 0.53 to 0.58 (Fig. 10a–f). Overall, both models capture the LH gradient across the study domain, with higher LH observed in the southern region and lower LH in the northern region. Similarly, both the reanalysis data and the models show a higher SH in the northern region and lower SH in the south. A systematic underestimation of LH (ranging between 22-35 W m$^{-2}$) and overestimation of SH (averaging 21-31 W m$^{-2}$) are evident in both WRF-ELM and WRF-CTSM. The observed evaporative fraction ranges from 0.6 to 0.8 in most vegetated grids; however, the corresponding simulated evaporative fraction is approximately 0.6. This evaluation further confirms that our models tend to underestimate LH fluxes while overestimating SH fluxes. These biases may be largely attributed to the surface parameters uncertainties used in the current simulations, such as LAI or roughness length. These parameters have not been thoroughly calibrated in coupled E3SM simulations focusing on the Great Lakes region."

[Figure]

Figure 10 (a-c) Spatial distribution of latent heat in (a) ERA5-Land (b) WRF-ELM, and (c) WRF-CTSM; (d-f) Spatial distribution of sensible heat in (d) ERA5-Land (e) WRF-ELM, and (f) WRF-CTSM; (g-h) Comparison of evaporative ratio between (g) WRF-ELM and ERA5-Land and (h) WRF-CTSM and ERA5-Land over the natural vegetation grids.

---

## Author Comment (AC2)

We sincerely appreciate the referees for their valuable and insightful comments on our manuscript. The feedback is instrumental in enhancing the quality and clarity of our research. These comments are not only valuable but also serve as a critical resource for improving various aspects of our article, including methodology, data interpretation, and overall presentation. We have taken each comment seriously and conducted a thorough review of our manuscript to ensure that we comprehensively address all concerns raised by the referees.

This response document provides a detailed account of the changes implemented in relation to each specific comment from the referee. For ease of reference, referee comments are presented in black, while authors' responses are provided in dark blue. The revised manuscript is highlighted in blue. The line numbers correspond to the revised version of the manuscript (track changed).

Reviewer #2

This study introduces the development of the U.S. Department of Energy's Energy Exascale Earth System Model (E3SM) Land Model (ELM) with Weather Research and Forecasting (WRF) Model coupled by a modified Lightweight Infrastructure for Land-Atmosphere Coupling (LILAC) framework (WRF-ELM v1.0). It is well-written and easy to understand. The method is publishable to improve the performance of land-atmosphere interaction in the land surface model regarding land heterogeneity. However, it will need major and minor revisions before it is considered for publication. This is because the authors made ESMF coupler (LILAC) but did not provide any improved performance of the model. Please see the following comments:

General/ Major Comments:

1) Land-atmosphere interactions (Atmosphere-land interaction in the title): How much does WRF-ELM improve to represent land-atmosphere interaction? Figures and tables in the paper show that WRF-ELM and WRF-CTSM have similar performances (surface sensible heat flux looks improved). The authors show figures of energy fluxes/temperature (land) and precipitation (atmosphere) but do not include information about land-atmosphere interaction.

R: Thank you for your insightful comment. Our understanding of land-atmosphere interaction aligns with Dickinson (1995), who describes the basic elements as the exchanges of moisture and energy between the land surface and the atmosphere. In the original manuscript, we have comprehensively evaluated latent heat flux (LH) and sensible heat flux (SH) from both WRF simulations and reanalysis, finding that both WRF models tend to overestimate . Additionally, we assessed skin temperature and precipitation, which directly reflect surface energy fluxes and atmospheric responses, respectively. To further investigate land-atmosphere interaction, in the revised manuscript, we have evaluated the evaporative fraction (LH/(LH+SH)), a critical indicator of land-atmosphere coupling strength, reflecting how surface energy is partitioned between sensible and latent heat fluxes. For most grids, we found the WRF models tend to underestimate LH while overestimate SH. The observed evaporative fraction ranges from 0.6 to 0.8 while, the corresponding simulated evaporative fraction is approximately 0.6.

We emphasize that the primary purpose of developing WRF-ELM is to build a flexible modeling framework that can seamlessly incorporate future advancements of ELM, such as heterogeneous land surfaces, dynamic land use, and plant hydraulics within the WRF modeling system. The E3SM land-atmosphere coupled simulations must be run globally and are highly computationally expensive at high resolutions. WRF-ELM, in contrast, allows us to investigate the impacts of new ELM features on regional land-atmosphere interactions using fewer computational resources and with a variety of parameterizations and schemes for atmospheric physical processes.

Here, we want to clarify that the comparison against WRF-CTSM and observations is not intended to demonstrate the superior performance of WRF-ELM but rather to ensure that the WRF-ELM coupling is functioning as intended. Our results indicate that WRF-ELM effectively captures key processes involved in land-atmosphere interactions at the regional scale, despite the absence of parameter calibration specific to the Great Lakes region. We have incorporated this discussion into the paragraphs below and added it to the manuscript on Lines 403-414 and Lines 255-257.

"We evaluated the simulated LH and SH fluxes from the WRF model simulations against ERA5-Land reanalysis data. The spatial correlation coefficients (CORR) range from 0.53 to 0.58 (Fig. 10a–f). Overall, both models capture the LH gradient across the study domain, with higher LH observed in the southern region and lower LH in the northern region. Similarly, both the reanalysis data and the models show a higher SH in the northern region and lower SH in the south. A systematic underestimation of LH (ranging between 22-35 W m$^{-2}$) and overestimation of SH (averaging 21-31 W m$^{-2}$) are evident in both WRF-ELM and WRF-CTSM. The observed evaporative fraction ranges from 0.6 to 0.8 in most vegetated grids; however, the corresponding simulated evaporative fraction is 0.4-0.6. This evaluation further confirms that our models tend to underestimate LH fluxes while overestimating SH fluxes. These biases may be largely attributed to the surface parameters uncertainties used in the current simulations, such as LAI, roughness length, or the btran parameterization, which describes the effect of soil moisture stress on stomatal conductance. These parameters have not been thoroughly calibrated in coupled E3SM simulations focusing on the Great Lakes region."

"We emphasize that the comparison against WRF-CTSM is not intended to demonstrate the superior performance of WRF-ELM but to show that the newly developed WRF-ELM performs comparably well to WRF-CTSM, one of the most advanced and sophisticated land surface models."

[Figure]

Figure 10 (a-c) Spatial distribution of latent heat in (a) ERA5-Land (b) WRF-ELM, and (c) WRF-CTSM; (d-f) Spatial distribution of sensible heat in (d) ERA5-Land (e) WRF-ELM, and (f) WRF-CTSM; (g-h) Comparison of evaporative ratio between (g) WRF-ELM and ERA5-Land and (h) WRF-CTSM and ERA5-Land over the natural vegetation grids.

Reference:

Dickinson, R. E.: Land-atmosphere interaction, Reviews of geophysics, 33, 917-922, 1995.

2) Heterogeneous Land Use:

Table 3 and figure 11 are made by forest regions from AmeriFlux. It is hard to generalize that the improvement in the model within only forest regions and the time series and biases of AmeriFlux are not matched with both models.

R: We appreciate the reviewer's comments regarding the challenges of generalizing model improvements based solely on forest regions and the discrepancies between AmeriFlux site observations regarding temporal series. Comparing regional model simulations with site-level observations remains a consistent difficulty for regional models due to the inherent scale mismatch between point observations and grid-based simulations. Additionally, temporal variations over the three-month period examined in this study further complicate the comparison, as neither model adequately captures these short-term dynamics. Our major conclusion for Fig.

11 is that the mean states of LH are better captured in WRF-ELM compared to WRF-CTSM. The above discussion has been added the revised manuscript (Lines 415-422).

"A further comparison of daily LH values from six AmeriFlux sites over deciduous broadleaf forests is illustrated in Fig. 11. WRF-ELM exhibits a smaller bias in reproducing the magnitude of LH than WRF-CTSM; however, neither model captures the temporal variations well. Comparing regional model simulations with site-level observations remains a consistent difficulty due to the inherent scale mismatch between point observations and grid-based simulations. Additionally, since we examined a relatively short period without interannual variability or seasonal cycles, the temporal variations of surface energy are mostly related to the simulation of cloud and precipitation variations, which are among the most uncertain parts of regional climate simulations."

3) Urban heat island (UHI):

It is suspicious that figure 9 is related to UHI, urban physics. It looks like more depending on the location by lake physics. Urban locations are closer than crop ones from the lake. For the maximum temperature from the diurnal cycle, urban regions are cooler than crop regions . It would not be the proper representation of UHI. It would be better to write the representation of lake breezes on urbanization in models, as mentioned in Wang et al. (2023), authors' reference. FYI, ASOS sites also show that urbans are cooler than crops.

R: Thanks for your comment. We agree with the reviewer that the temperature and moisture differences between urban and crop are not only due to these land types, but also due to the lake effect especially during the summer season when lake breeze is present during noon to afternoon. The daytime T2 and RH difference can be due to the lake breeze cooling effect as documented in Wang et al. 2023, making the maximum temperature not a good indicator of UHI representation but a better representation of lake breeze. Both ASOS and WRF capture this difference too. However, we emphasize that the higher T2 and lower RH during nighttime (as seen from ASOS), specifically minimum T2 and RH, reflects the urban heat island and urban dry land effect. Both WRF simulations capture the warmer T2 in urban area than crop during nighttime but the RH differences are not well represented. Possible biases arise from urban and crop module in both models and model setup need to be further investigated. We have revised our manuscript below and not limit the discussion to urban heat island but also consider the lake breeze cooling effect (Lines 372-391):

"We use ASOS sites to investigate the representation of urban and lake effects on air temperature and relative humidity over the metropolitan area, emphasizing the interaction between the urban heat island (UHI; Rizwan et al., 2008) and lake breeze in WRF-ELM and WRF-CTSM. Six urban sites along the west coast of Lake Michigan were selected, paired with six adjacent crop sites as reference points (Fig. 9a). Compared to the rural crop sites, the urban sites exhibit higher minimum Ta during the night, as urban areas retain more heat during the daytime and gradually release after sunset. During late morning to noon, the lake breeze tends to cool urban air, resulting in a lower daily maximum Ta than observed in crop areas (Wang et al., 2023). In the afternoon, urban sites show a more gradual decline in Ta compared to rural sites, driven by the cumulative heating effect of solar radiation absorption and the heat release by urban materials throughout

the day (Soltani and Sharifi, 2017). This characteristic of urban areas leads to a smaller diurnal temperature range of 7.0 K, compared to a 9.0 K range over crop sites (Figs. 9b-c). The UDI effect is also evident in 2m RH observations from ASOS, with urban areas showing lower RH values at night (Figs. 9d-e).

Both WRF-ELM and WRF-CTSM capture the warmer nighttime Ta due to the UHI effect and the cooler daytime Ta caused by the lake breeze over urban sites, adequately reproducing the smaller diurnal range. WRF simulations, particularly WRF-ELM, reasonably capture urban RH at night, but both models underestimate RH over crop areas, so the UDI is not well captured in the simulations. Notably, WRF-ELM generally exhibits smaller biases in both Ta and RH compared to WRF-CTSM (Fig. 9). However, both models systematically overestimate T2 and underestimate RH in both urban and crop areas, suggesting a persistent warm and dry bias need to be further investigated in the ELM and CTSM component."

Minor Comments:

1) Figure1: There are lots of abbreviations. It would be needed to make a supplementary table to explain them.

R: We appreciate your suggestion. An inserted table has been added below Fig. 1 which include abbreviations of all models.

2) Figure 4 and MPI: If geographic domain is 7 x 4 gird cells, does MPI not work due to P0 is not equal to P1?

R: If geographic domain is 7 x 4 grid cells, P0 will process on 4 x 4 ELM grid cells while P1 will work on 3 x 4 ELM grid cells. MPI does not inherently prevent processors from handling different numbers of grid cells. Instead, it can handle unequal domain decomposition.

3) Table 2: NCEP Stabe IV dataset missed.

R: It has been added in the revised version.

4) I may think that figure 7 and L346 – 347 are not necessary. I am not sure about the relationship between spatial smoothing and land heterogeneity.

R: The model configuration used in WRF-CTSM does not consider subgrid heterogeneity and adopts only a dominant vegetation type to represent surface biogeophysical properties. For example, the model assigns 100% grass fraction in the southern part and 100% tree fraction in the northern part of the zoomed domain, leading to an abrupt change in vegetation types and resulting in an apparent temperature contrast due to differing surface properties (Fig. 7). In contrast, the configuration we adopt WRF-ELM incorporates a mixture of plant functional types

within each grid cell, which allows for a more gradual transition of vegetation fractions from south to north, leading to smoother temperature gradients across the domain.

5)  Are soil properties updated in WRF-ELM?

R: In the current WRF-ELM, we use temporally static soil properties, for example, soil organic matter, soil color, and soil texture from ELM2/CLM5 default parameters derived from Batjes (2006)

Reference:

Batjes, N. H.: ISRIC-WISE derived soil properties on a 5 by 5 arc-minutes global grid, Report 2006/02, ISRIC, http://www.isric.org (last access: 6 June 2023), 2006.

Other comments:

1) Title: "Atmosphere-land" but "Land-atmosphere" in the context.

R: Thank you for pointing this out. It has been uniformed to "Land-atmosphere" throughout the manuscript.

2) Figure 2 and L262: Plant functional type (PFT)

R: The full name has been provided in Fig. 2 caption and L267